# Microstructure Evolution and Mechanical Properties of a Wire-Arc Additive Manufactured Austenitic Stainless Steel: Effect of Processing Parameter

**DOI:** 10.3390/ma14071681

**Published:** 2021-03-29

**Authors:** Ping Long, Dongxu Wen, Jie Min, Zhizhen Zheng, Jianjun Li, Yanxing Liu

**Affiliations:** 1State Key Laboratory of Materials Processing and Die & Mould Technology, School of Materials Science and Engineering, Huazhong University of Science and Technology, Wuhan 430074, China; lnpin@hust.edu.cn (P.L.); minjie@hust.edu.cn (J.M.); zzz@hust.edu.cn (Z.Z.); jianjun@hust.edu.cn (J.L.); 2College of Mechanical Engineering, Dongguan University of Technology, Dongguan 523808, China; talkingbird@dgut.edu.cn

**Keywords:** austenitic stainless steel, microstructure, mechanical properties, wire arc additive manufacturing, fracture characteristic

## Abstract

Two single track multi-layer walls with linear energy inputs (LEIs) of 219 and 590 J/mm were deposited by cold metal transfer-based wire arc additive manufacturing system. Combined with the X-ray diffraction technique, scanning electron microscope and uniaxial tensile tests, the influences of LEI and cooling rate (CR) on the microstructure evolution, mechanical properties and fracture mechanisms of the studied steel are analyzed. It is observed that the microstructures of the studied steel are mainly composed of δ-ferrite and austenite dendrites. σ phase is formed on the δferrite–austenite interface under low CR. Meanwhile, the primary dendrites’ spacing decreases with the decrease in LEI or the increase in CR, and the maximal primary dendrites’ spacing is 32 μm. The values of elongation to fracture roughly decline with the decrease in LEI or the increase in CR, but the variations of ultimate tensile strength and yield stress show an opposite trend. In addition, the mesoscopic damages in the studied steel under low LEI are mainly caused by the coalescence of pores. While under high LEI, the cracks are induced by the dislocations piling up around δ-ferrite.

## 1. Introduction

Generally, the complex components are hardly integrally manufactured by traditional forming processes (such as casting and forging) and need to be divided into several subcomponents [1,2]. Due to the high design freedom and material utilization, additive manufacturing (AM) technologies are widely applied to fabricate or repair the complex metallic components by the local delivery of metal wire or powder [3,4,5]. According to the heat sources, the AM technologies are classified into wire arc additive manufacturing (WAAM) [6,7], laser powder-based fusion (LPBF) [8,9], and electron beam melting (EBM) [10,11]. LPBF and EBM are commonly used to form components with high dimensional accuracy [12]. However, the applications of EBM and LPBF need to be kept in enclosed environments [13]. Therefore, the component sizes are greatly influenced by the spacing of the enclosed working chamber.

Recently, larger components were expected to be fabricated through additive manufacturing technologies [14,15]. Due to the low equipment cost and high deposition rate, great attention has been paid to WAAM technology [16]. The traditional commercial welding power source can be directly applied in the WAAM process. Moreover, the WAAM enjoys a potentially unlimited component size in open working environments [17,18]. However, there are still some disadvantages for the application of WAAM. During WAAM, the processing parameters, including wire feed speed, scanning speed and shielding gas flow, significantly affect the size, thermodynamic and dynamic properties of the molten pool [19,20]. For the large temperature gradient and cycling reheating, the preferred layer sizes, fine microstructures as well as homogeneous elements distributions are hardly obtained [21,22]. After cooling, the microstructures of the components are constituted by plenty of columnar dendrites [23]. The inhomogeneous microstructure and anisotropy effect would further influence the mechanical properties of additive manufactured components [24]. The relatively poor dimensional accuracy is another unfavorable impact for the wide application of WAAM [25,26]. For overcoming the above shortcomings, many researches focusing on dimensional and microstructure control of components were conducted for the WAAM process [27,28,29]. Dai et al. [30] studied the slicing and scanning strategies of the WAAM process for manufacturing a multi-directional pipe joint, and successfully controlled the dimensional errors within ±1 mm. Yang et al. [31] analyzed the influences of inter-layer cooling time on the forming appearance of parts by an infrared camera. Xiong et al. [32] investigated the influences of processing parameters on the surface quality of additive manufactured layers and developed a passive vision sensor system to control the layer height. Sun et al. [33] discussed the mechanical properties and microstructure evolution of Inconel 625 superalloy component deposited by WAAM. Considering the multiple thermal cycle effects, Lin et al. [34] employed two WAAM methods (pulsed and continuous plasma arc additive manufacturing) to fabricate Ti-6Al-4V alloy components and revealed the relationship between mechanical properties and microstructures. Guo et al. [35] addressed the effects of pulse frequency on microstructures and the tensile properties of AZ31 magnesium alloy. Additionally, some studies were concerned with the surface morphology, mechanical properties and microstructure of metals, including aluminum alloys [36,37], nickel-based alloys [38,39], titanium alloys [40,41], steels [42,43], as well as intermetallic compounds [44,45] manufactured by WAAM.

Due to the excellent machinability, weldability, mechanical properties and high corrosion resistance, the austenitic stainless steel is widely applied as the structural material used in corrosive environments [46,47,48], such as pipelines, containers and heat exchangers in nuclear, biomedicine and petrochemical fields. With the development of industrial equipment, the size, precision and complexity of parts increases. Compared with traditional casting and forging, WAAM technology is a simple forming process with high production efficiency and material utilization rate [49]. In addition, with the development of WAAM technology, the mechanical properties of 316L WAAM-formed parts are higher than casting parts, and sometimes even reach the level of forging parts [50].

During the AM process, the high cooling rate aggravates the inhomogeneous elements distribution. δ-ferrite cannot totally transform into austenite in a short cooling period [4]. The microstructures of AM-fabricated components mainly consist of residual δ-ferrite and austenite [51]. Yadollahi et al. [52] clarified the influences of inter-layer time and thermal history on the mechanical properties and microstructures of austenitic stainless steel. Guo et al. [53] investigated the anisotropy of the direct laser deposition (DLD)-processed 316L stainless steel. Ziętala et al. [54] paid attention to the corrosion resistance properties of 316L stainless steel. In order to improve the corrosion resistance and mechanical properties of 316L stainless steel, Chen et al. [55] optimized the heat treatment process to obtain the optimal volume fraction of σ phase and δ-ferrite. Yan et al. [56] investigated the creep–fatigue properties of 316L stainless steel and established a model to predict the life span of studied steel. Miranda et al. [57] established models to clarify the relationship between the processing parameters and mechanical properties of selective laser-melted 316L stainless steel. Xu et al. [58] discussed the influences of annealing treatment on the microstructures of 316LN stainless steel in the temperature range of 600–1000 °C. Wu et al. [59] observed the microstructural evolution of 316LN stainless steel in situ under various cooling rates by a confocal scanning laser microscope. Xiao et al. [60] optimized the arc welding heat input to obtain the preferred cryogenic toughness of 316LN stainless steel. Dai et al. [61] revealed the effects of heat treatment on low-temperature toughness and the microstructure evolution of an austenitic stainless steel.

There are many processing parameters in the WAAM process. Essentially, most parameters, such as scanning speed and wire feeding speed, affect the material properties by changing the energy input and cooling rate of the molten pool. This article directly considers the influence of energy input and cooling rate on the material performance, so that it can help one to understand the microstructure evolution and phase transition mechanism in the WAAM process. In this article, multi-layer 316L stainless steel parts were fabricated by the cold metal transfer (CMT)-based WAAM system, and the effects of the linear energy input (LEI) and cooling rate (CR) on the microstructure evolution, mechanical properties and fracture mechanisms of the studied steel were analyzed in detail.

## 2. Materials and Experiments

The ER316L stainless steel wire with a diameter of 1.2 mm used in this investigation was provided by the TIAUVD company. The chemical compositions are shown in Table 1. The forged 316LN stainless steel baseplate with a size of 200 × 100 × 30 mm^3^ was grinded and washed in alcohol solution prior to the WAAM process. Two single track multi-layer parts were fabricated by the cold metal transfer (CMT)-based WAAM system, which is shown in Figure 1. Figure 2 shows the schematic of the CMT-based WAAM system. The system is composed of four parts, namely shielding gas, a CMT heat source (CMT 4000 advanced), a robot arm (KUKA KR30-3HA) and an industry personal computer. The gas metal arc was struck between the wire electrode and thin wall. The CMT program during the WAAM process is CMT. As the welding torch moved, wire was melted, and the part was deposited layer by layer. The height of the AM part is 32 mm, and the length is 150 mm. During experiment, the scanning direction of the adjacent layers was reversed. The time interval between two layers was set to 3 min. The flow rate of the protecting gas (80% Ar + 20% CO_2_) was 25 L/min, and the inner diameter of the gas nozzle was 15 mm. Table 2 shows the processing parameters of WAAM used in this study. The low and high LEI groups represent that the studied steel deposited under the LEI of 219 and 590 J mm^−1^, respectively.

Generally, different regions of the AM parts undergo different CRs during the WAAM process. In the bottom region of the part, a great deal of heat input is quickly absorbed by the baseplate, and the CR is fast. With the distance away from the baseplate increasing, the effects of heat accumulation and thermal cycle are apparent. Thus, the CR decreases with the increased distance away from the baseplate. In this study, the longitudinal tensile samples were cut from different regions of the AM parts along the scanning direction. The distances from the baseplate to the center line of the bottom, middle and top samples were measured as 5, 15 and 25 mm, respectively. To further investigate the anisotropic effects of the AM parts, the vertical tensile samples were cut along the building direction. The detailed sample position within the AM part and the dimension of sample are shown in Figure 3. Three longitudinal and six vertical tensile samples were made per parameter. Uniaxial tensile tests were carried out on a mechanical testing machine (AG-IC 100kN, Shimadzu Co., Kyoto, Japan) at a strain rate of 2 × 10^−3^ s^−1^. After fracture, both the transverse and longitudinal sections of samples were observed by SEM (Sirion200, FEI Co., Eindhoven, The Netherlands). 

The optical microscope (OM) was adopted to characterize microstructures. The chemical etchant was a mixture solution consisting of 5 mL of HNO_3_ and 15 mL of HCl. The etching time was around 15 s. X-ray diffraction (XRD) tests (D8 Advance, Broker Co., Karlsruhe, Germany) were conducted with a scan rate of 0.02°/s from 30° to 100°.

## 3. Results and Discussion

### 3.1. Microstructure Evolution

As mentioned above, different regions of the AM parts undergo different CRs during the WAAM process. Due to heat accumulation, the CR declines with the increased layers. Figure 4 illustrates the XRD patterns of the AM parts under different LEIs and CRs. It is clear that the diffraction peaks at 41°, 50°, 74°and 90° are well matched with austenite (JCPDS card No. 31-0619). The peaks at 42°, 62° and 81° demonstrate the existence of δ-ferrite. Therefore, the AM parts under different LEIs and low/medium CRs is comprised by δ-ferrite and austenite. The high diffraction peaks of {200} and {220} plane of austenite demonstrates that <100> and <110> are the two favorable growth orientations of the dendrites. However, the diffraction peak of (45°) σ phase (CrFe) is only available in the AM part under high CR. This manifests that σ phase only precipitate under low CR during WAAM. Generally, σ phase precipitate at the ferrite–austenite interface once the temperature stays in the range of 600–900 °C [62,63]. On the other hand, σ phase would dissolve in the austenite matrix when the temperature is above 1050 °C [64]. For the studied steel under high CR, the duration of temperature within 600–900 °C is short. The time is insufficient for the formation of σ phase. With the increase in distance from the baseplate, the effect of heat accumulation is obvious, and the CR decreases. Therefore, enough time is provided for the precipitation of σ phase in the studied steel under medium and high CRs. However, due to the thermal cycle caused by successive layer deposition, the duration of temperature above 1050 °C is long enough for the resolution of σ phase. Therefore, only austenite and δ-ferrite can be found in the AM parts under a medium CR. In the top regions of the AM parts, the duration of temperature exceeding 1050 °C is short, owing to the effect of heat transfer with air. Thus, the σ phase finally survives when CR is low.

Figure 5 shows the macro-morphologies of the top region within the AM parts. The layer widths are 8.2 ± 0.7 mm and 11.4 ± 1.2 mm for the AM parts under low and high LEIs, respectively. A relatively large layer width indicates a small temperature gradient under high LEI. Moreover, a host of columnar dendrites grow along the building direction, as shown in Figure 5b.

The microstructures of the AM parts under different LEIs and CRs are shown in Figure 6. The lathy and skeletal δ-ferrite are observed in the AM parts under medium and high CRs, but only lathy δ-ferrite is available under low CRs. Generally, with the decrease in temperature, the phase transformation follows [65]:L→L+δ-ferrite→L+δ-ferrite+austenite→δ-ferrite+austenite→austenite
where L denotes liquid phase. 

During WAAM, owing to the fast CR, the solid phase transformation from δ-ferrite to austenite is non-equilibrium and incomplete, and the skeletal or lathy δ-ferrite survives. Table 3 shows the values of the residual δ-ferrite contents and primary dendrites’ spacing. The contents of residual δ-ferrite under different WAAM processing parameters are evaluated by ImageJ software. For each condition, 10 different metallographic images are selected, and the average values of residual δ-ferrite are obtained. Under low LEI, the spacing of primary dendrites is evaluated as 27, 21 and 18 μm for the AM part with low, medium and high CRs, respectively. With LEI increasing, the spacing of primary dendrites raise to 32, 27 and 23 μm at the low, medium and high CRs, respectively. It is obvious that the primary dendrites’ spacing increases with the decrease in CR. However, the δ-ferrite contents follow an opposite tendency. When LEI is low, the contents of δ-ferrite are 14.5%, 16.2% and 17.3% with the cooling rate decreasing. Under high LEI, the δ-ferrite contents range from 13.2% to 17.7%. In the vicinity of the baseplate, the deposited heat quickly dissipates through conduction. The grain growth is not apparent. With the progress of WAAM, the average temperature of the AM parts increases and the CR decreases. Thus, there is enough time for the columnar dendrites’ growth, which further results in the primary dendrites’ spacing increasing. Meanwhile, numerous δ-ferrite dissolves into the austenite matrix. In consequence, the variation trend of the δ-ferrite content is opposite to that of the primary dendrites’ spacing for the studied steel with different CRs.

Generally, the primary dendrites’ spacing is greatly affected by the nucleation rate and the holding time suitable for grain growth. The nucleation rate mainly depends on the undercooling degree at the solid–liquid interface during solidification. A great undercooling degree accelerates the nucleation of the columnar dendrites. The interface undercooling degree is calculated by the following formulas [66]:(1)$$V_{s}=V\cos\theta$$
(2)$$\Delta T_{k}=\frac{V_{s}}{\gamma }$$
where $V_{s}$ is the moving speed of solid–liquid interface, *V* is the welding speed, *θ* is the angle between *V* and $V_{s}$. $\Delta T_{k}$ is the undercooling degree of the interface front, and γ is the material constant related to the adopted material and local acoustic speed.

According to Equation (2), $\Delta T_{k}\;$depends on the moving speed of the interface and is correlated to the welding speed in this study. A fast welding speed leads to a large undercooling degree. The welding speed of the AM part under low LEI is relatively small. Thus, the undercooling degree is large, and the nucleation rate is high.

During WAAM, the grain growth model is expressed by [67]: (3)$$g^{2}=K_{1}\tau \exp(-\frac{Q}{RT_{p}})+{g_{0}}^{2}$$
(4)$$\tau =\frac{q}{2\pi \lambda e(T_{p}-T_{0})}$$
where$\;g_{0}$ and *g* are the initial and ultimate grain size, respectively. *Q* denotes the thermal activation energy, R is the molar gas constant, and $k_{1}$ is material constant. $T_{0}$ and $T_{p}$ are the preheating and peak temperature, respectively. λ is the thermal conductivity, τ is the time suitable for grain growth, and q is the linear energy input.

In this model, the initial grain size is assumed as a constant. The time suitable for grain growth is mainly determined by q and the difference between $T_{0}$ and $T_{p}$. The peak temperature $T_{p}$ approximately reaches the melting temperature of materials during WAAM. Due to the effect of heat accumulation, $T_{0}\;$gradually increases with the increase in additive layers. Therefore, the time suitable for grain growth gradually increases with the increased layers. On the other hand, the differences between the preheating temperature $T_{0}$ and the peak temperature $T_{p}$ are not apparent in the same regions of the AM part. Thus, a suitable  τ for grain growth is mainly related to LEI and increases with the LEI increasing. In conclusion, the primary dendrites’ spacing of the studied steel increases with the increase in LEI or the decrease in CR.

### 3.2. Mechanical Properties

Figure 7 depicts the engineering stress–strain relationships of the studied steel. Obviously, the mechanical properties are significantly affected by LEI and CR. In detail, Figure 8 shows the yield strength (YS), ultimate tensile strength (UTS) and elongation to fracture of the studied steel under different LEIs and CRs, and the detailed values are listed in Table 4. Under a low LEI, the UTSes are 533, 553 and 573 MPa for the studied steel with low, medium and high CRs, respectively. With the increase in LEI, the UTSes decrease to 521, 530 and 553 MPa for the studied steel under low, medium and high CRs, respectively. The UTS and YS both show an increasing trend with the increase in CR or the decrease in LEI. As mentioned in Section 3.1, the columnar dendrites are gradually refined with the decrease in LEI or the increase in CR. The decrease in the primary dendrite spacing provides numerous grain boundaries, which considerably hinders the movement of dislocations. Therefore, the YS and UTS are preferred for the AM parts under low LEI or high CR. 

From Figure 7, it is found that the UTSes and YSes of the vertical tensile samples are relatively low compared with the longitudinal samples under different LEIs. This is mainly resulted from two aspects. Firstly, the columnar dendrites grow along the building direction, which brings about a significant decrease in grain boundaries in the vertical direction. Thus, the grain boundary strengthening effect is diminished, compared with that in the longitudinal direction. On the other hand, the inter-layer interface is likely to induce some defects such as pores, which further weakens the tensile properties. Therefore, the longitudinal tensile samples have preferred UTSes and YSes compared with vertical samples. Additionally, the elongation to fracture relatively follows an increased trend with the increase in LEI or CR, as shown in Figure 8.

### 3.3. Fracture Morphologies

Figure 9 shows the longitudinal sectional morphologies of the vertical tensile samples under different LEIs. The sectional shrinkage rates are evaluated as 48.8% and 54.1% for the studied steel under low and high LEIs, respectively. Generally, a large sectional shrinkage rate indicates the material with an excellent plasticity. Figure 9b,c,e,f are the magnified graphs of regions marked as 1, 2, 3 and 4 in Figure 9a,d, respectively. Obviously, δ-ferrite is thick in the studied steel under low LEI, and is elongated along the tensile direction during the progress of tension, as shown in Figure 9b,c. Moreover, pores and cracks are observed, and the number of pores is relatively large under low LEI. During the solidification of the additive layer, the fast cooling rate prevents the escape of gas from the liquid metal, and numerous pores are formed. It also can be found from Figure 9b that the cracks are mainly caused by the coalescence of pores for the studied steel under low LEI. With the increase in LEI, a small number of pores exist on the longitudinal sectional morphologies of the tensile samples, as shown in Figure 9e. The coalescence of pores is not apparent with the progress of the tension. Meanwhile, several cracks occur along the tensile direction, as shown in Figure 9f. This is mainly attributed to hindering the movement of dislocations by δ-ferrite during the tensile deformation. A great number of dislocations pile up around δ-ferrite, resulting in the stress concentration. Once the critical bonding stress of the interface is exceeded, cracks occur. Therefore, the fracture of the studied steel under high LEI results from the disbanding between δ-ferrite and austenite matrix. 

Figure 10 shows the fracture surfaces of the vertical tensile samples under different LEIs. The fracture morphologies are covered with numerous equiaxed dimples and tearing edges. This indicates that the formation and coalescence of micro-voids lead to the final fracture of the studied steel. Besides, large dimples and micro-cracks are formed by the coalescence of dimples, which severely deteriorates the plasticity of the studied steel. From Figure 10a, it is found that numerous dimples distribute on the fracture surfaces for the studied steel under low LEI. Several cleavage facets caused by the coalescence of pores also exist on the fracture surfaces. It is found from Figure 10c that the dimples of the studied steel under high LEI are relatively deep compared to that in Figure 10a. In general, deep dimples mean that the dimples have experienced large deformation before the final fracture. Therefore, the studied steel under high LEI has preferred plastic deformation capacity. This conclusion coincides well with the variation of elongation for the studied steel under different LHIs, as shown in Figure 8. Figure 10b,d depicts the high-resolution graphs of the circled regions in Figure 10a,c, respectively. In Figure 10d, the number of the broken δ-ferrite particles and dimples increase, compared with those in Figure 10b. The δ-ferrite hinders the movement of dislocations, resulting in dislocations piling up at the interface between austenite matrix and δ-ferrite. Once the induced stress reaches a threshold, the micro-voids occur. In addition, the serpentine sliding characteristic becomes distinct under high LEI (Figure 10d). The serpentine sliding is the new form of sliding process occurring after the formation of dimples, which indicates a preferable plasticity [68,69,70]. 

Figure 11 shows the fracture surfaces of the longitudinal tensile samples under different LEIs and CRs. Obviously, plenty of dimples and tearing edges are formed under different LEIs and CRs, indicating the final fracture of the studied steel is caused by the coalescence of micro-voids. Under low LEI, the dimple size gradually reduces when increasing the CR, as shown in Figure 11a–c. These phenomena are highly related to the respective δ-ferrite contents. As listed in Table 3, the content of δ-ferrite increases with the CR increasing. During the tension, the micro-voids nucleate and coalesce around δ-ferrite due to the stress concentration. Therefore, lots of dimples are easily formed for the studied steel under high CR. This also interprets the increment of the UTS and TS with the increase in CR. 

From Figure 11b,f, it is found that cracks are also formed due to the coalescence of dimples on the fracture surface for the studied steel under a medium CR. Besides, the cleavage facets exist on the fracture morphologies of the longitudinal tensile samples under low LEI (Figure 11c). In the same region of the AM parts, the dimples under high LEI are slightly enlarged in the width and depth (Figure 11d–f), compared with those under low LEI. Therefore, the studied steel under high LEI enjoys a preferred plasticity, compared with that under low LEI. This conclusion is well consistent with the above analyses in fracture morphologies of the vertical tensile samples. 

## 4. Conclusions

Numerous researches have been made to investigate the influences of WAAM process parameters on the material properties. Most of the parameters affect the microstructure and performance of the material by changing the energy input and CR of the molten pool. In order to understand the mechanism of the WAAM process, the present study directly investigates the influences of LEI and CR on microstructure evolution and mechanical properties of the wire-arc additive manufactured 316L stainless steel. Some insights are summed as:(1)The microstructures of the studied steel are mainly composed of δ-ferrite and austenite dendrites. σ phases are formed on the δ-ferrite–austenite interface under low CR. The contents of δ-ferrite show an increasing trend with the decrease in CR. However, the effects of LEI on the δ-ferrite content are not apparent.(2)Under low LEI, the UTSes are 533, 553 and 573 MPa for the studied steel with low, medium and high CRs, respectively. With the increase in LEI, the UTSes decrease to 521, 530 and 553 MPa. The UTS and YS both show an increasing trend with the increase in CR or the decrease in LEI. However, the variation of elongation to fracture shows an opposite trend.(3)Numerous dimples and tearing edges are distributed on the fracture morphologies of the studied steel, indicating that the main fracture mechanism is the micro-voids-induced ductile fracture. The dimples are relatively deep for the studied steel under high LEI compared with that under low LEI, and the dimple size decreases with the CR increasing. For the studied steel under low LEI, the cracks are induced by the coalescence of pores. However, the cracks in the studied steel under high LEI are result from the dislocations piling up around δ-ferrite.

## Figures and Tables

**Figure 1 materials-14-01681-f001:**
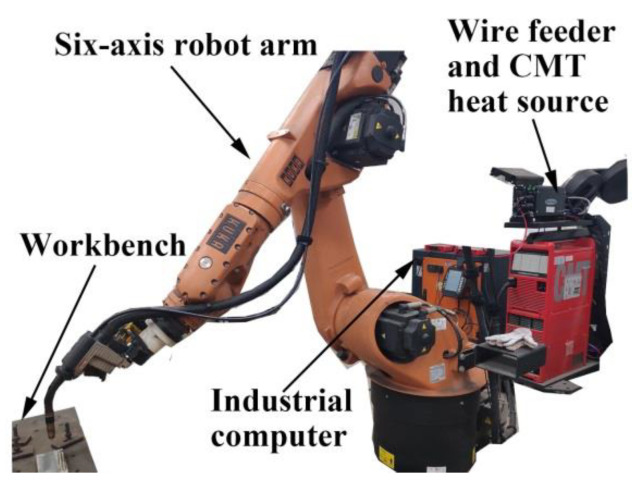
Cold metal transfer (CMT)-based wire arc additive manufacturing (WAAM) system.

**Figure 2 materials-14-01681-f002:**
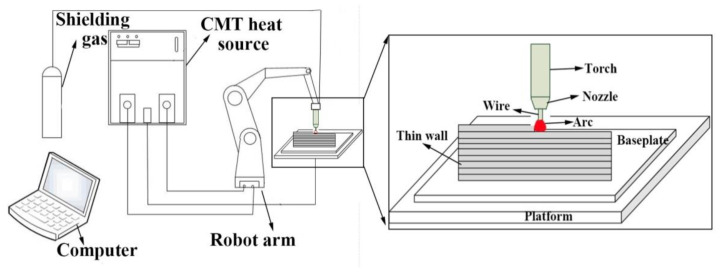
Schematic description of the CMT-based WAAM system.

**Figure 3 materials-14-01681-f003:**
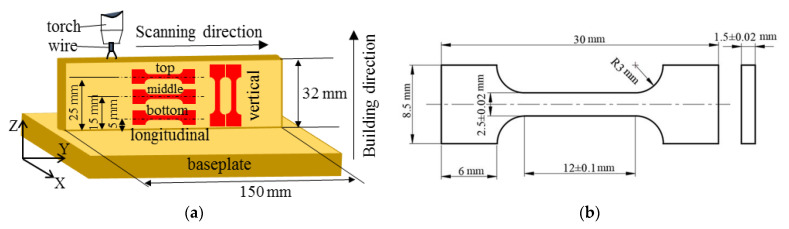
Position and dimension of tensile samples: (**a**) sample position within the additive manufacturing (AM) part (Y and Z denote the scanning direction and building direction, respectively); (**b**) detailed dimension of the tensile sample.

**Figure 4 materials-14-01681-f004:**
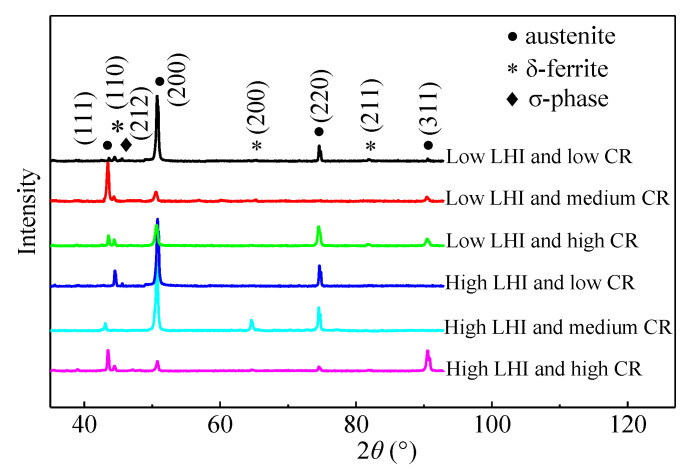
XRD patterns of the AM parts under different LEIs and cooling rates (CRs).

**Figure 5 materials-14-01681-f005:**
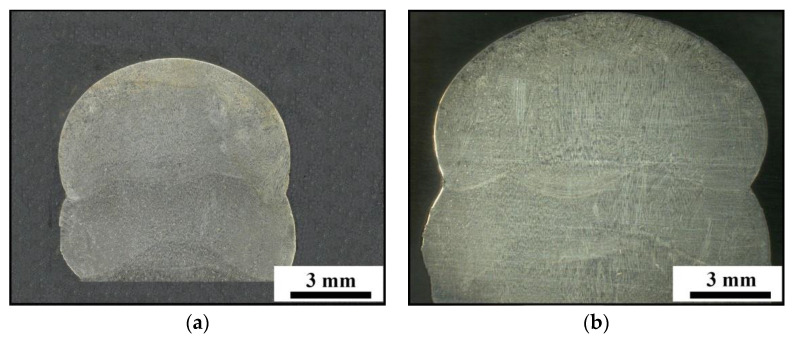
Macro-morphologies of the top region within the AM parts under: (**a**) low LEI; (**b**) high LEI.

**Figure 6 materials-14-01681-f006:**
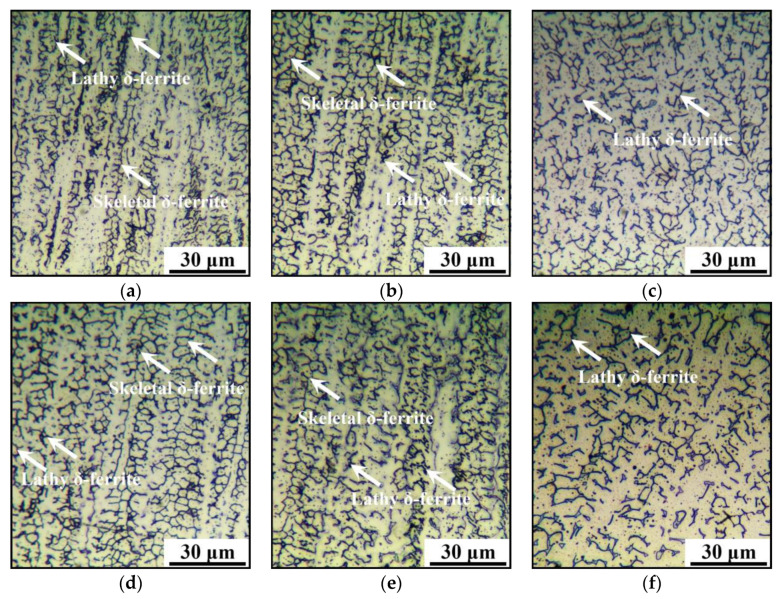
Microstructures of the AM parts with: (**a**) low LEI and high CR; (**b**) low LEI and medium CR; (**c**) low LEI and low CR; (**d**) high LEI and high CR; (**e**) high LEI and medium CR; (**f**) high LEI and low CR.

**Figure 7 materials-14-01681-f007:**
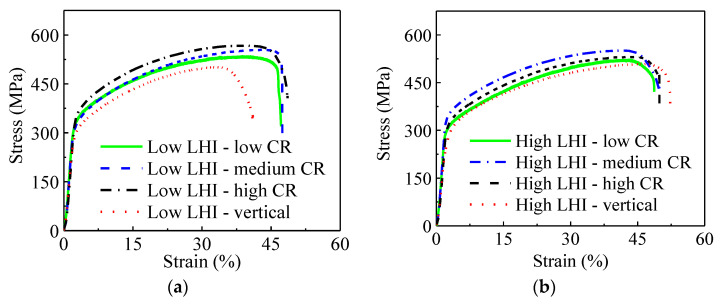
Engineering stress–strain relationships of the studied steel under: (**a**) low LEI, (**b**) high LEI.

**Figure 8 materials-14-01681-f008:**
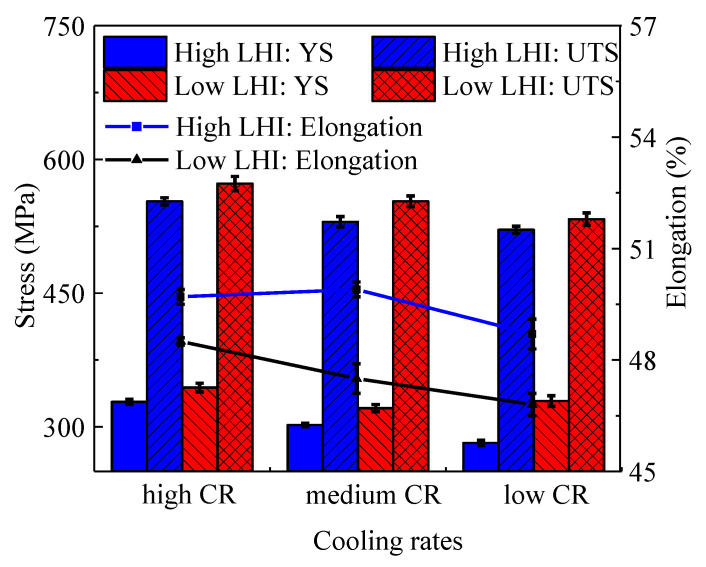
Histogram of mechanical properties of longitudinal tensile samples under different LEIs.

**Figure 9 materials-14-01681-f009:**
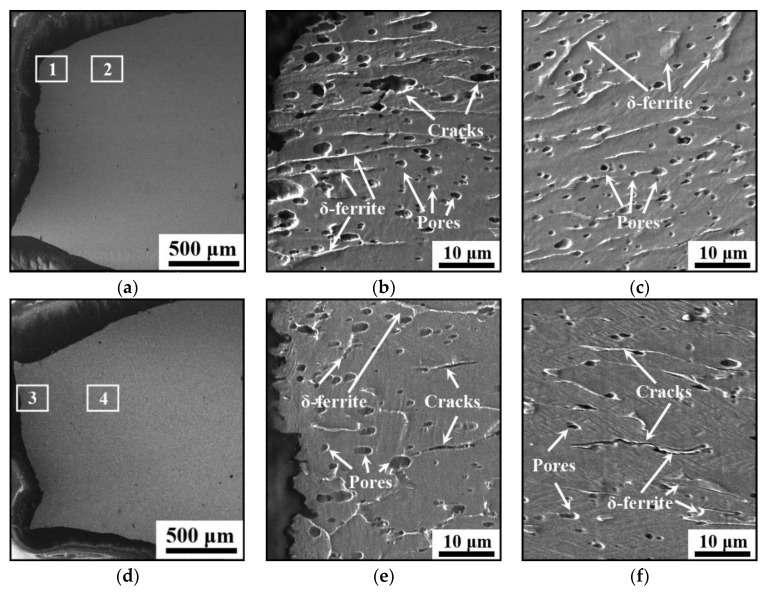
Longitudinal sectional morphologies of the vertical tensile samples under: (**a**–**c**) low LEI; (**d**–**f**) high LEI. (**b**,**c**,**e**,**f**) are the magnified graphs of regions marked 1, 2, 3, 4 in (**a**,**d**).

**Figure 10 materials-14-01681-f010:**
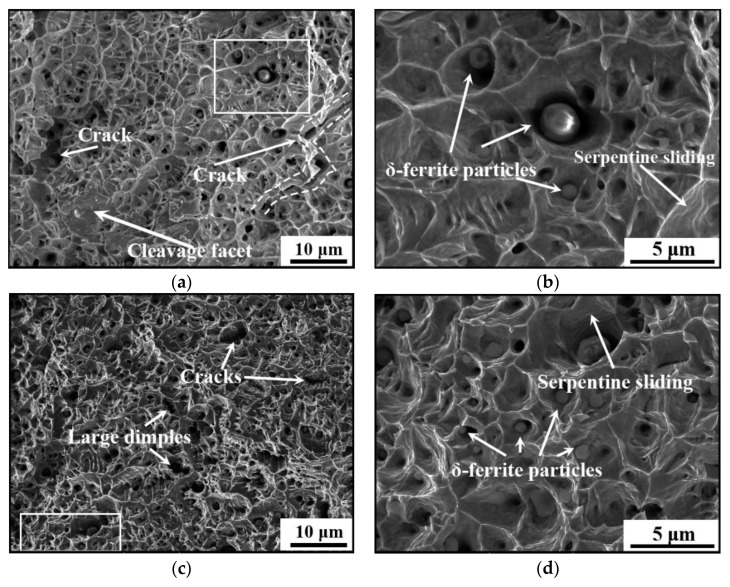
Fracture surfaces of the vertical tensile samples under: (**a**,**b**) low LEI; and (**c**,**d**) high LEI.

**Figure 11 materials-14-01681-f011:**
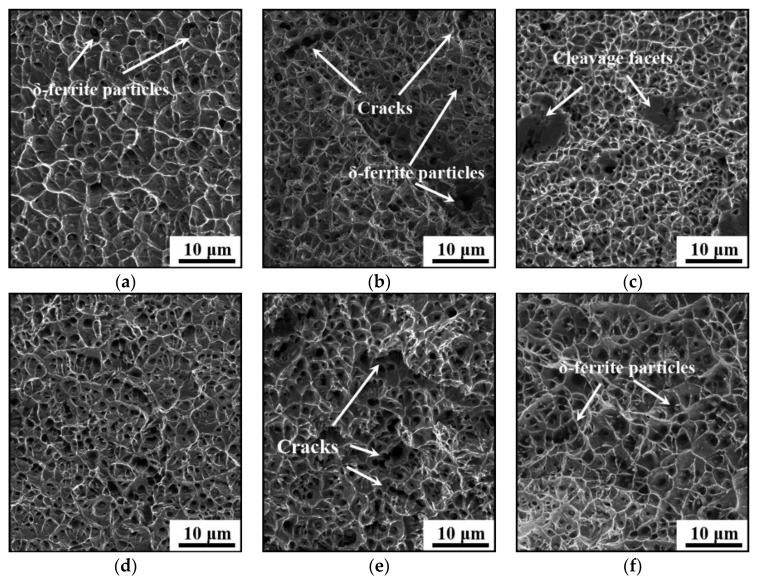
Fracture surfaces of the longitudinal tensile samples under (**a**) low LEI and low CR; (**b**) low LEI and medium CR; (**c**) low LEI and high CR; (**d**) high LEI and low CR; (**e**) high LEI and medium CR; (**f**) high LEI and high CR.

**Table 1 materials-14-01681-t001:** Chemical composition of ER316L stainless steel wire.

Element	Cr	Ni	Mo	Mn	Si	C	S	P	N	Fe
wt.%	18.39	12.5	2.25	1.69	0.81	0.02	0.015	0.015	0.013	balance

**Table 2 materials-14-01681-t002:** Processing parameters of WAAM used in this study. LEI: linear energy input.

Group	Wire Feed Speed (mm × s^−1^)	Travel Speed (mm × s^−1^)	Voltage (V)	Current (A)	LEI (J × mm)
Low LEI	50	8	14.4	122	219
High LEI	83	5	18.2	162	590

**Table 3 materials-14-01681-t003:** δ-ferrite contents and primary dendrites spacing of the AM parts under different LEIs and CRs.

Microstructure Characteristics	The Studied Steel					
	Low LEI- High CR	Low LEI- Medium CR	Low LEI- Low CR	High LEI- High CR	High LEI- Medium CR	High LEI- Low CR
Primary dendrites spacing (μm)	18	21	27	23	27	32
δ-ferrite contents (%)	17.3	16.2	14.5	17.7	16.2	13.2

**Table 4 materials-14-01681-t004:** Summary of the mechanical properties of the studied steel under different LEIs and CRs.

Mechanical Properties	The Studied Steel					
	Low LEI- High CR	Low LEI- Medium CR	Low LEI- Low CR	High LEI- High CR	High LEI- Medium CR	High LEI- Low CR
Yield stress (MPa)	344±5	321±4	329±6	328±3	302±2	282±3
Ultimate tensile stress (MPa)	573±8	553±6	533±7	553±4	530±6	521±4
Elongation (%)	48.5±0.4	47.5±0.1	46.8±0.3	49.7±0.2	49.9±0.2	48.7±0.4

## Data Availability

The study did not report any data.

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
