# Peer review of "Microstructure Evolution and Mechanical Properties of a Wire-Arc Additive Manufactured Austenitic Stainless Steel: Effect of Processing Parameter"

_materials, 2021, doi:10.3390/ma14071681_

Round 1

Reviewer 1 Report

There are some weaknesses through the manuscript which need improvement. Therefore, the submitted manuscript cannot be accepted for publication in this form, but it has a chance of acceptance after a major revision. My comments and suggestions are as follows:

1- Abstract gives information on the main feature of the performed study, but some details about the specimens and printing process be added.

2- Authors must clarify necessity of the performed research. Objectives of the study must be clearly mentioned.

3- The literature study must be enriched. In this respect, authors must read and refer to the following papers: (a) effects of printing parameters: https://doi.org/10.1016/j.prostr.2020.10.083  (b) metal 3D printing: https://doi.org/10.1016/j.repl.2020.12.007

4- It would be nice, if authors could add some figures (real or schematic) to show printed specimens.

5- The main reference of each formula must be cited. Moreover, each parameters in equations must be introduced. Please double check this issue.

6- Since this manuscript presents an experimental study, adding several figures to show experimental test conditions is a necessity.

7- The presented curves (e.g., Fig. 5) must be prepared in MATLAB environment.

8- In its language layer, the manuscript should be considered for English language editing. There are sentences which have to be rewritten.

9- The conclusion must be more than just a summary of the manuscript. List of references must be updated based on the proposed papers. Please provide all changes by red color in the revised version.

Reviewer 2 Report

In this article, authors present the effect of processing parameters during WAAM (cooling rate and linear heat input) on the microstructure and tensile properties of austenic stainless steel grade 316L.

Three different cooling rates were tested in combination with two linear heat inputs. Various cooling rates were essentially achieved by taking samples from three different locations on the deposited wall (bottom, middle and top part). Two linear heat inputs were essentially a combination of low wire feed speed + high travel speed and high wire feed speed + low travel speed.

The results are in my opinion interesting and of a great importance in future WAAM understanding and development. There are, however, some key issues to be addressed before publication.

  1. In the introduction, I suggest the following papers to be reviewed to make sure the literature overview is up to date:
    • Lam et al. Adaptive process control implementation of wire arc additive manufacturing for thin-walled components with overhang features. International Journal of Advanced Manufacturing Technology 2020;108:1061–71. https://doi.org/10.1007/s00170-019-04737-4
    • Oliveira et al. Revisiting fundamental welding concepts to improve additive manufacturing: From theory to practice. Progress in Materials Science 2020;107:100590. https://doi.org/10.1016/j.pmatsci.2019.100590
    • ŠÄŤetinec et al. In-process path replanning and online layer height control through deposition arc current for gas metal arc based additive manufacturing. Journal of Manufacturing Processes 2021;64:1169–79. https://doi.org/https://doi.org/10.1016/j.jmapro.2021.02.038
  2.  In the chapter Materials and experiments authors should specify exact CMT machine and CMT program that was used during the research (CMT advanced, CMT Advanced Pulse, etc)
  3. In the same chapter shielding gas flow rate is given but not the gas nozzle diameter.
  4. In the same chapter (line 118): Authors should specify how many  horizontal samples were made per parameter? Similarly, how many vertical samples -figure 1 suggests two?
  5. Chapter Results and discussion: Figure three would present the difference better if both images would be made using the same magnification.
  6. In the conclusions section (line 358) "The dimples are relatively deep..." The term "relatively" is not appropriate and should at least be elaborated when the dimples are considered "deep" or compared to another example
  7. Finally, authors should carefully check the manuscript for spelling errors and a number of expressions/terms should be changed to more appropriate ones. Such as: 
    • line 108: "... two layers was settled as 3 minutes..." should be changed to "... two layers was set to 3 minutes ..."
    • line 106: "the CMT system was equipped on a 6-axis ..." should be changed to "6-axis KUKA robot was equipped with ..."
    • line 189: the term "inadequate" is not correct in this case
    • line 302: "... high LEI enjoys ..." The term "enjoys" should be changed to a more suitable one

Reviewer 3 Report

The reviewer comments of the paper «Microstructure evolution and mechanical properties of a wire-arc additive manufactured austenitic stainless steel: Effect of processing parameter»

- Reviewer

The authors presented an article «Microstructure evolution and mechanical properties of a wire-arc additive manufactured austenitic stainless steel: Effect of processing parameter». However, there are several points in the article that require further explanation.

Comment 1:

Introduction.

It is useful to add a short paragraph for industrial applications of ER316L stainless steel. What are the properties of the material? Show the advantages and disadvantages in comparison with the classical method of obtaining.

At the end of the introduction, add a clear and concise purpose of the article.

Comment 2:

  1. Materials and experiments

Instead of a description in the text of the article, it would be better and more visual to add a table with the chemical composition of the material under study.

What is the hardness of the material and how was it measured?

Comment 3:

  1. Results and discussion

For better visualization and clarity, redraw the figures 2, 5, 6 in color.

Figure 9 and its description are best placed in the previous section before conclusions.

Comment 4:

It will be useful to add a section of Nomenclature in which to sign all the physical quantities and abbreviations encountered in the article. There are many physical quantities in the text and such a section will help to find the description of the necessary element.

For example,

ε             : Strain

CR         : Cooling rate

etc.

Comment 5:

Conclusions.

It is necessary to more clearly show the novelty of the article and the advantages of the proposed method. What is the difference from previous work in this area? Show practical relevance. What is the difference from other researchers?

The article is interesting and written at a good scientific level. Authors should carefully study the comments and make improvements to the article step by step. After major changes can an article be considered for publication in the "Materials".

Round 2

Reviewer 1 Report

The paper has been improved and corresponding modifications have been conducted. In my opinion, the current version can be considered for publication.

Author Response

Thanks for your suggestion.

Reviewer 2 Report

Authors have addressed all of the recommendations. Just one thing more: 

In the line 122: The diameter of the gas nozzle (gas lens) surely was not just 1.4 mm, as typical nozzle diameter is  ~15-25 mm?

Before publication, authors should carefully read the final manuscript and eliminate any spelling errors that still remain

Author Response

Thanks for your suggestion. So sorry for this mistake. We have further checked the nozzle diameter, and the inner diameter of gas nozzle is measured by 15 mm. We also checked and revised the manuscript to eliminate the spelling errors. Thanks.

Reviewer 3 Report

The authors have improved the article according to the comments. The article can now be published.

Author Response

Thanks for your suggestion.